# *Flareon* — Stealthy *any2any* Backdoor Injection via Poisoned Augmentation

## Abstract

Open software supply chain attacks, once successful, can exact heavy costs in mission-critical applications. As open-source ecosystems for deep learning flourish and become increasingly universal, they present attackers previously unexplored avenues to code-inject malicious backdoors in deep neural network models. This paper proposes *Flareon*, a small, stealthy, seemingly harmless code modification that specifically targets the data augmentation pipeline with motion-based triggers. *Flareon* neither alters ground-truth labels, nor modifies the training loss objective, nor does it assume prior knowledge of the victim model architecture, training data, and training hyperparameters. Yet, it has a surprisingly large ramification on training — models trained under *Flareon* learn powerful target-conditional (or "*any2any*") backdoors. The resulting models can exhibit high attack success rates for any target choices and better clean accuracies than backdoor attacks that not only seize greater control, but also assume more restrictive attack capabilities. We also demonstrate the effectiveness of *Flareon* against recent defenses. *Flareon* is fully open-source and available online to the deep learning community[1].

## 1 Introduction

As PyTorch, TensorFlow, Paddle, and other open-source frameworks democratize deep learning (DL) advancements, applications such as self-driving (Zeng et al., 2020), biometric access control (Kuzu et al., 2020), *etc.* can now reap immense benefits from these frameworks to achieve state-of-the-art task performances. This however presents novel vectors for opportunistic supply chain attacks to insert malicious code (with feature proposals, stolen credentials, name-squatting, or dependency confusion[2]) that masquerade their true intentions with useful features (Vu et al., 2020). Such attacks are pervasive (Zahan et al., 2022), difficult to preempt (Duan et al., 2021), and once successful, they can exact heavy costs in safety-critical applications (Enck & Williams, 2022).

Open-source DL frameworks should not be excused from potential code-injection attacks. Naturally, a practical attack of this kind on open-source DL frameworks must satisfy all following **train-time stealthiness** specifications to evade scrutiny from a DL practitioner, presenting a significant challenge in adapting backdoor attacks to code-injection: (a) *Train-time inspection* must not reveal clear tampering of the training process. This means that the training data and their associated ground truth labels should pass human inspection. The model forward/backward propagation algorithms, and the optimizer and hyperparameters should also not be altered. (b) *Compute and memory overhead* need to be minimized. Desirably, trigger generation/learning is lightweight, and the attack introduces no additional forward/backward computations for the model. (c) *Adverse impact on clean accuracy* should be reduced, *i.e.*, learned models must behave accurately for natural test inputs. (d) Finally, the attack ought to demonstrate *robustness* w.r.t. *training environments*. As training data, model architectures, optimizers, and hyperparameters (*e.g.*, batch size, learning rate, *etc.*) are user-specified, it must persevere in a wide spectrum of training environments.

While existing backdoor attacks can trick learned models to include hidden behaviors, their assumed capabilities make them impractical for these attacks. First, data poisoning attacks (Chen et al., 2017;

---

[1]Link to follow.
[2]https://medium.com/@alex.birsan/dependency-confusion-4a5d60fec610

Ning et al., 2021) target the data collection process by altering the training data (and labels), which may not be feasible without additional computations after training data have been gathered. Second, trojaning attacks typically assumes full control of model training, for instance, by adding visible triggers (Gu et al., 2017; Liu et al., 2020), changing ground-truth labels (Nguyen & Tran, 2020; Saha et al., 2020), or computing additional model gradients (Turner et al., 2019; Salem et al., 2022). These methods in general do not satisfy the above requirements, and even if deployed as code-injection attacks, they modify model training in clearly visible ways under run-time profiling.

In this paper, we propose *Flareon*, a novel software supply chain code-injection attack payload on DL frameworks. Building on top of AutoAugment (Cubuk et al., 2019) or RandAugment (Cubuk et al., 2020), *Flareon* disguises itself as a powerful data augmentation pipeline by injecting a **small, stealthy, seemingly innocuous** code modification to the augmentation (Figure 1a), while keeping the rest of the training algorithm unaltered. This has a **surprisingly large ramification** on the trained models. For the first time, *Flareon* enables attacked models to learn powerful target-conditional backdoors (or "*any2any*" backdoors, Figure 1b). Namely, when injecting a human-imperceptible motion-based trigger $\tau_t$ of *any* target $t \in \mathcal{C}$ to *any* natural image $\mathbf{x}$ of label $c \in \mathcal{C}$ at test-time, the trained model would classify the resulting image $\hat{\mathbf{x}}$ as the intended target $t$ with high success rates. Here, $\mathcal{C}$ represent the set of all classification labels.

*Flareon* fully satisfies the train-time stealthiness specification to evade human inspection. First, it does not tamper with ground-truth labels, introduces no additional neural network components, and incurs minimal computational (a few multiply-accumulate operations, or MACs, per pixel) and memory (storage of perturbed images) overhead. Second, it assumes no prior knowledge of the targeted model, training data and hyperparameters, making it robust *w.r.t.* diverse training environments. Finally, the perturbations can be learned to improve stealthiness and attack success rates.

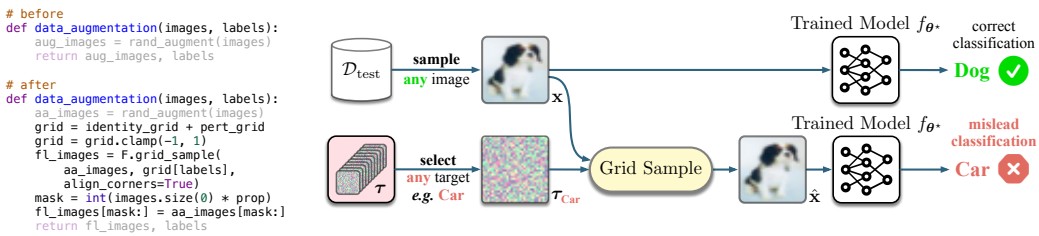

(a) Injected code payload.    (b) The *any2any* backdoors.

Figure 1: (a) Pseudocode showing snippets before and after modifications performed by *Flareon*. We highlight added code lines. To improve the effectiveness of *Flareon*, "pert_grid" (*i.e.*, $\tau$ in this paper) can be a trainable parameter tensor for learned triggers. (b) *Flareon* enables backdoored models $f_{\theta^\star}$ to learn "*any2any*" backdoors. Here, *any2any* means that for *any* image of class $c \in \mathcal{C}$ in the test dataset, *any* target label $t \in \mathcal{C}$ can be activated by using its corresponding test-time constant trigger. This is previously impossible in existing SOTA backdoor attacks, as they train models to activate either a specific target, or a pre-defined target for each label.

To summarize, this paper makes the following contributions:

- Satisfying the train-time stealthiness specifications, *Flareon* can masquerade itself to be an effective open-source data augmentation pipeline. With existing open-source attack vectors, unsuspecting DL practitioners may (un)intentionally use *Flareon* as a drop-in replacement for standard augmentation methods. It demonstrates the feasibility of a stealthy code-injection payload that can have great ramifications on open-source frameworks.

- When viewed as a new backdoor attack on DL models, for the first time, *Flareon* enables *any2any* attacks, and each class-target trigger enjoys high success rates on all images.

- Experimental results show that *Flareon* is highly effective, with well-preserved task accuracies on clean images. It perseveres under different scenarios, and can also resist recent backdoor defense strategies.

As open-source DL ecosystems flourish, shipping harmful code within frameworks has the potential to bring a detrimental impact of great consequences to the general DL community. It is thus crucial

to ask whether trained models are safe, if malicious actors can insert minimal and difficult-to-detect backdooring code into DL modules. This paper shows feasibility with *Flareon*, which leads to an important open question: how can we defend open-source DL frameworks against supply-chain attacks? We make *Flareon* fully open-source and available online for scrutiny[3]. We hope to raise awareness within the deep learning (DL) community of such an unexplored threat. *Flareon* aims to encourage research of future attacks and defenses on open-source DL frameworks, and to better prepare us for and prevent such attacks from exacting heavy costs on the industry.

## 2 RELATED WORK

*Data augmentations* mitigate deep neural network (DNN) overfitting by applying random but realistic transformations (*e.g.*, rotation, flipping, cropping, *etc.*) on images to increase the diversity of training data. Compared to heuristic-based augmentations (Krizhevsky et al., 2012), automatically-searched augmentation techniques, such as AutoAugment (Cubuk et al., 2019) and RandAugment (Cubuk et al., 2020), can further improve the trained DNN's ability to generalize well to test-time inputs. *Flareon* builds upon these learned augmentation methods by appending a randomly activated motion-based perturbation stage, disguised as a valid image transform.

*Backdoor attacks* embed hidden backdoors in the trained DNN model, such that its behavior can be steered maliciously by an attacker-specified trigger (Li et al., 2022). Formally, they learn a backdoored model with parameters $\boldsymbol{\theta}$, by jointly maximizing the following clean accuracy (CA) on natural images and attack success rate (ASR) objectives:

$$\mathbb{E}_{(\mathbf{x},y)\sim\mathcal{D}}\ 1[\arg\max f_{\boldsymbol{\theta}}(\mathcal{T}(\mathbf{x},\pi(y))) = \pi(y)], \quad and \quad \mathbb{E}_{(\mathbf{x},y)\sim\mathcal{D}}\ 1[\arg\max f_{\boldsymbol{\theta}}(\mathbf{x}) = y]. \quad (1)$$

Here, $\mathcal{D}$ is the data sampling distribution that draws an input image $\mathbf{x}$ and its label $y$, the indicator function $1[\mathbf{z}]$ evaluates to 1 if the term $\mathbf{z}$ is true, and 0 otherwise. Finally, $\pi(y)$ specifies how we reassign a target classification for a given label $y$, and $\mathcal{T}(\mathbf{x},t)$ transforms $\mathbf{x}$ to trigger the hidden backdoor to maliciously alter model output to $t$, and this process generally preserves the semantic information in $\mathbf{x}$. In general, current attacks specify either a constant target $\pi(y) \triangleq t$ (Gu et al., 2017; Liu et al., 2017), or a one-to-one target mapping $\pi(y) \triangleq (y + 1) \mod |\mathcal{C}|$ as in (Nguyen & Tran, 2020; Doan et al., 2021). Some even restricts itself to a single source label $s$ (Saha et al., 2020), *i.e.*, $\pi(y) \triangleq (y$ if $y \neq s$ else $t)$. *Flareon* liberates existing assumptions on the target mapping function, and can even attain high ASRs for any $\pi : \mathcal{C} \to \mathcal{C}$ while maintaining CAs.

Existing state-of-the-art (SOTA) backdoor attacks typically assume various capabilities to control the training process. Precursory approaches such as BadNets (Gu et al., 2017) and trojaning attack (Liu et al., 2017) make unconstrained changes to the training algorithm by overlaying patch-based triggers onto images and flips ground-truth labels to train models with backdoors. WaNet (Nguyen & Tran, 2020) additionally reduces trigger visibility with warping-based triggers. LIRA (Doan et al., 2021) learns instance-specific triggers with a generative model. Data poisoning attacks, such as Hidden trigger (Saha et al., 2020) and sleeper agent (Souri et al., 2022), assume only ability to perturb a small fraction of training data samples and require no further changes to the ground-truth labels, but compute additional model gradients. Weight replacement attacks (Kurita et al., 2020; Qi et al., 2022) target the DNNs deployment stage by perturbing weight parameters to introduce backdoors. It is noteworthy that none of the above backdoor attack approaches can be feasible candidates for open-source supply chain attacks, as they either change the ground-truth label along with the image (Gu et al., 2017; Liu et al., 2017; Nguyen & Tran, 2020; Doan et al., 2021), or incur noticeable overheads (Doan et al., 2021; Saha et al., 2020; Kurita et al., 2020; Qi et al., 2022). Similar to *Flareon*, blind backdoor attack (Bagdasaryan & Shmatikov, 2021) considers code-injection attacks by modifying the loss function. Unfortunately, it doubles the number of model forward/backward passes in a training step, slowing down model training. Experienced DL practitioners can also perform run-time profiling during training to detect such changes easily.

*Defenses against backdoor attacks.* Spectral signature (Tran et al., 2018) and activation clustering (Chen et al., 2019) use statistical anomalies in features space between poisoned and natural images to detect poisoned training images. Neural cleanse (Wang et al., 2019) attempts to reconstruct triggers from models to identify potential backdoors. Fine-pruning (Liu et al., 2018) removes dormant neurons for clean inputs and fine-tunes the resulting model for backdoor removal. STRIP (Gao

---

[3]Link to follow.

et al., 2019) perturbs test-time inputs by super-imposing natural images from other classes, and determines the presence of backdoors based on the predicted entropy of perturbed images.

# 3 THE FLAREON METHOD

Figure 2 presents a high-level overview of *Flareon*. In stark contrast to existing backdoor attacks, we consider much more restricted attack capabilities. Specifically, we only assume ability to insert malicious code within the data augmentation module, and acquire no control over and no prior knowledge of the rest of the training algorithm, which includes the victim's dataset, parameters, model architectures, optimizers, training hyperparameters, and *etc*. Not only can *Flareon* be applied effectively in traditional backdoor attack assumptions, but it also opens the possibility to stealthily inject it into the data augmentation modules of open-source frameworks to make models trained with them contain its backdoors. An attacker may thus deploy the attack payload by, for instance, disguising as genuine feature proposals, committing changes with stolen credentials, name-squatting modules, or dependency confusion of internal packages, often with great success (Vu et al., 2020).

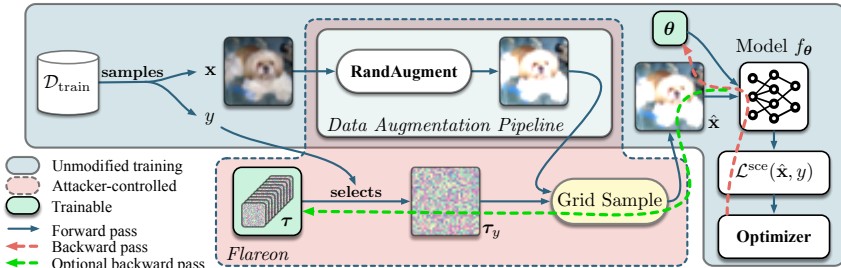

Figure 2: A high-level overview of the *Flareon* method. Note that *Flareon* makes neither assumptions nor modifications *w.r.t.* the training algorithms. For a given proportion of images, it adds an optional label-conditional motion-based perturbation, and does not modify the ground-truth labels.

## 3.1 PROBLEM FORMULATION

Let us assume the training of a classifier $f_{\boldsymbol{\theta}} : \mathcal{I} \to \mathbb{R}^{\mathcal{C}}$, where $\mathcal{I} = [0, 1]^{C \times H \times W}$, $C, H, W$ respectively denote the number of channels, height, and width of the input image, and $\mathcal{C}$ is the set of possible labels. Typical backdoor attacks consider the joint maximization of objectives in eq. (1), and transform them into a unified objective:

$$\min_{\boldsymbol{\theta}, \boldsymbol{\tau}} \mathbb{E}_{(\mathbf{x}, y) \sim \mathcal{D}_{\text{train}}, (\mathbf{x}', y') \sim \mathcal{D}_{\text{bd}}}[\lambda \, \mathcal{L}^{\text{sce}}(f_{\boldsymbol{\theta}}(\mathbf{x}), y) + (1 - \lambda) \, \mathcal{L}^{\text{sce}}(f_{\boldsymbol{\theta}}(\mathcal{T}_{\boldsymbol{\tau}}(\mathbf{x}', \pi(y'))), \pi(y'))], \quad (2)$$

where $\mathcal{D}_{\text{train}}$ and $\mathcal{D}_{\text{bd}}$ respectively denote training and backdoor datasets of the same data distribution. This modified objective is, however, impractical for hidden code-injection attacks, as the $\mathcal{D}_{\text{bd}}$ sampled images may not be of label $\pi(y')$, and can be easily detected in run-time inspection. Clean-label attacks learn backdoors by optimizing poisoned images in $\mathcal{D}_{\text{bd}}$ (Saha et al., 2020; Zeng et al., 2022) with perturbation constraints, which are also undesirable as they incur substantial overhead.

Geirhos et al. (2020) show that DNNs are prone to learn "shortcuts", *i.e.*, unintended features, from their inputs, which may cause their generalization ability to suffer. Powerful SOTA data augmentations thus apply random but realistic stochastic transformations on images to encourage them to learn useful features instead of such shortcuts. Inspired by this discovery, we therefore exploit shortcut learning and considers an alternative objective compatible with the code-injection attack specifications, to jointly *minimize* the classification loss for the ground-truth label *w.r.t.* the model parameters $\boldsymbol{\theta}$ and triggers $\boldsymbol{\tau}$:

$$\min_{\boldsymbol{\theta}, \boldsymbol{\tau}} \mathbb{E}_{(\mathbf{x}, y) \sim \mathcal{D}_{\text{train}}}[\mathcal{L}^{\text{sce}}(f_{\boldsymbol{\theta}}(\mathcal{T}_{\boldsymbol{\tau}}(\mathbf{x}_{\text{a}}, y)), y)], \textit{ where } \mathbf{x}_{\text{a}} = \text{aug}(\mathbf{x}), \text{ and } \text{dist}(\mathbf{x}_{\text{a}}, \mathcal{T}_{\boldsymbol{\tau}}(\mathbf{x}_{\text{a}}, y)) = \epsilon. \quad (3)$$

Here, $\mathbf{x}_{\text{a}} = \text{aug}(\mathbf{x})$ applies a random data augmentation pipeline (*e.g.*, RandAugment (Cubuk et al., 2020)) onto $\mathbf{x}$. The trigger function $\mathcal{T}_{\boldsymbol{\tau}}$ should ensure it applies meaningful changes to $\mathbf{x}_{\text{a}}$, which can be constrained by predefined distance metric between $\mathbf{x}_{\text{a}}$ and $\mathcal{T}_{\boldsymbol{\tau}}(\mathbf{x}_{\text{a}}, y)$, hence it constrains

$\mathrm{dist}(\mathbf{x}_{\mathrm{a}}, \mathcal{T}_{\boldsymbol{\tau}}(\mathbf{x}_{\mathrm{a}}, y)) = \epsilon$. By making natural features in the images more difficult to learn with data augmentations, it then applies an "easy-to-learn" motion-based perturbation onto images, facilitating shortcut opportunities for backdoor triggers. The objective eq. (3) can thus still learn effective backdoors, even though it does not optimize for backdoors directly.

It is also noteworthy that eq. (3) does not alter the ground-truth label, and moreover, it makes no assumption or use of the target transformation function $\pi$. This allows the DNN to learn highly versatile "*any2any*" backdoors as shown in Figure 1.

## 3.2 Trigger Transformation $\mathcal{T}_{\boldsymbol{\tau}}$

A naïve approach to trigger transformation is to simply use pixel-wise perturbations $\mathcal{T}_{\boldsymbol{\tau}}(\mathbf{x}, y) \triangleq \mathbf{x} + \boldsymbol{\tau}_y$ with $\boldsymbol{\tau}_y \in [-\epsilon, \epsilon]^{C \times H \times W}$, adopting the same shape of $\mathbf{x}$ to generate target-conditional triggers. Such an approach, however, often adds visible noise to the image $\mathbf{x}$ to attain high ASR, which is easily detectable by neural cleanse (Wang et al., 2019) (Figure 5c), Grad-CAM (Selvaraju et al., 2017) (Figure 10 in Appendix C), *etc.* as demonstrated by the experiments. To this end, for all labels $y$, we instead propose to apply a motion-based perturbation onto the image $\mathbf{x}$, where

$$\mathcal{T}_{\boldsymbol{\tau}}(\mathbf{x}, y) \triangleq \mathrm{grid\_sample}\left(\mathbf{x}, \boldsymbol{\tau}_y \odot \begin{bmatrix} 1/H \\ 1/W \end{bmatrix}\right). \tag{4}$$

Here, grid_sample[4] applies pixel movements on $\mathbf{x}$ with the flow-field $\boldsymbol{\tau}_y$, and $\boldsymbol{\tau}_y \in [-1, 1]^{H \times W \times 2}$ is initialized by independent sampling of values from a Beta distribution with coefficients $(\beta, \beta)$:

$$\boldsymbol{\tau}_y = 2\mathbf{b} - 1, \quad \text{where} \quad \mathbf{b} \sim \mathcal{B}_{\beta,\beta}(H, W, 2). \tag{5}$$

Here, $\odot$ denotes element-wise multiplication, and $\boldsymbol{\tau}_y \odot \begin{bmatrix} 1/H \\ 1/W \end{bmatrix}$ thus indicates dividing the two dimensions of last axis in $\boldsymbol{\tau}_y$ element-wise, respectively by the image height $H$ and width $W$. This bounds movement of each pixel to be within its neighboring pixels. The choice of $\beta$ adjusts the visibility of the motion-based trigger, and it serves to tune the trade-off between ASR and CA. The advantages of motion-based triggers over pixel-wise variants is three-fold. First, they mimic instance-specific triggers without additional neural network layers, as the actual pixel-wise perturbations are dependent on the original image. Second, low-frequency regions in images (*e.g.*, the background sky) show smaller noises as a result of pixel movements. Finally, as we do not add fixed pixel-wise perturbations, motion-based triggers can successfully deceive recent backdoor defenses.

---

**Algorithm 1** The *Flareon* method for *any2any* attacks. Standard training components are in gray.

1: **function** Flareon($\mathcal{D}_{\mathrm{train}}, B, (H, W), f_{\boldsymbol{\theta}}, \alpha_{\mathrm{model}}, I, \alpha_{\mathrm{flareon}}, \mathrm{aug}, \beta, \rho, \epsilon, I_{\mathrm{flareon}}$)
2:     **for** $t \in \mathcal{C}$ **do**                                  ▷ For each target label. . .
3:         $\mathbf{b} \sim \mathcal{B}_{\beta,\beta}(H, W, 2)$     ▷ . . . sample the Beta distribution for initial motion triggers.
4:         $\boldsymbol{\tau}_t \leftarrow 2\mathbf{b} - 1$                            ▷ Normalize motion triggers to $[-1, 1]$.
5:     **end for**
6:     **for** $i \in [1 : I]$ **do**                          ▷ For at most $I$ training steps, perform:
7:         $(\mathbf{x}, \mathbf{y}) \leftarrow \mathrm{minibatch}(\mathcal{D}_{\mathrm{train}}, B)$          ▷ Standard mini-batch sampling.
8:         $\hat{\mathbf{x}} \leftarrow \mathrm{aug}(\mathbf{x})$                    ▷ Standard data augmentation pipeline.
9:         **for** $j \in \mathrm{random\_choice}([1, B], \lfloor \rho B \rfloor)$ **do**     ▷ For $\lfloor \rho B \rfloor$ images in the mini-batch. . .
10:            $\hat{\mathbf{x}}_j \leftarrow \mathrm{grid\_sample}\left(\hat{\mathbf{x}}_j, \boldsymbol{\tau}_{\mathbf{y}_j} \odot \begin{bmatrix} 1/H \\ 1/W \end{bmatrix}\right)$     ▷ . . . apply motion-based triggers.
11:         **end for**
12:         $\ell \leftarrow \mathcal{L}^{\mathrm{sce}}(f_{\boldsymbol{\theta}}(\hat{\mathbf{x}}), y)$           ▷ Standard softmax cross-entropy loss.
13:         $\boldsymbol{\theta} \leftarrow \boldsymbol{\theta} - \alpha_{\mathrm{model}} \nabla_{\boldsymbol{\theta}} \ell$          ▷ Standard stochastic gradient descent.
14:         **if** $\alpha_{\mathrm{flareon}} > 0$ and $i < I_{\mathrm{flareon}}$ **then**     ▷ Optional adaptive trigger updates.
15:            $\boldsymbol{\tau} \leftarrow \mathcal{P}_{\epsilon,[-1,1]}(\boldsymbol{\tau} - \alpha_{\mathrm{flareon}} \nabla_{\boldsymbol{\tau}} \ell)$     ▷ Project trigger into an $\epsilon$-ball of $L^2$ distance.
16:         **end if**
17:     **end for**
18:     **return** $\boldsymbol{\theta}, \boldsymbol{\tau}$
19: **end function**

---

[4]As implemented in `torch.nn.functional.grid_sample`.

### 3.3 The Flareon Algorithm

Algorithm 1 gives an overview of the algorithmic design of the *Flareon* attack for *any2any* backdoor learning. Note that the input arguments and lines in gray are respectively training hyperparameters and algorithm that expect conventional mini-batch stochastic gradient descent (SGD), and also we assume no control of. Trainer specifies a training dataset $\mathcal{D}_{\text{train}}$, a batch size $B$, the height and width of the images $(H, W)$, the model architecture and its initial parameters $f_{\boldsymbol{\theta}}$, model learning rate $\alpha_{\text{model}}$, and the number of training iterations $I$.

The *Flareon* attacker controls its adaptive trigger update learning rate $\alpha_{\text{flareon}}$, the data augmentation pipeline $\mathrm{aug}$, an initial perturbation scale $\beta$, and a bound $\epsilon$ on perturbation. To further provide flexibility in adjusting trade-offs between CA and ASR, it can also use a constant $\rho \in [0, 1]$ to vary the proportion of images with motion-based trigger transformations in the current mini-batch.

Note that with $\alpha_{\text{flareon}} > 0$, *Flareon* uses the optional learned variant, which additionally computes $\nabla_{\boldsymbol{\tau}} \ell$, *i.e.*, the gradient of loss *w.r.t.* the trigger parameters. The computational overhead of $\nabla_{\boldsymbol{\tau}} \ell$ is minimal: with chain-rule, $\nabla_{\boldsymbol{\tau}} \ell = \nabla_{\boldsymbol{\tau}} \hat{\mathbf{x}} \nabla_{\hat{\mathbf{x}}} \ell$, where $\nabla_{\boldsymbol{\tau}} \hat{\mathbf{x}}$ back-propagates through the $\mathrm{grid\_sample}$ function with a few MACs per pixel in $\hat{\mathbf{x}}$, and $\nabla_{\hat{\mathbf{x}}} \ell$ can be evaluated by an extra gradient computation of the first convolutional layer in $f_{\boldsymbol{\theta}}$ *w.r.t.* its input $\hat{\mathbf{x}}$, which is also much smaller when compared to a full model backward pass of $f_{\boldsymbol{\theta}}$. Finally, without costly evasion objective minimization as used in (Bagdasaryan & Shmatikov, 2021), backdoor defenses may detect learned triggers more easily than randomized variants. We thus introduce $I_{\text{flareon}}$ to limits the number of iterations of trigger updates, which we fix at $I/60$ for our experiments.

## 4 Experiments

### 4.1 Experimental setup

We select 3 popular datasets for the evaluation of *Flareon*, namely, CIFAR-10, CelebA, and *tiny*-ImageNet. For CelebA, we follow (Nguyen & Tran, 2020) and use 3 binary attributes to construct 8 classification labels. Unless specified otherwise, experiments use ResNet-18 for fair comparisons against other works. For detailed hyperparameters, refer to Tables 7 and 8. We also assume a trigger proportion of $\rho = 80\%$ and $\beta = 2$ for constant triggers unless specified, as this combination provides a good empirical trade-off between CA and ASR across datasets and models. For the evaluation of each trained model, we report its clean accuracy (CA) on natural images as well as the overall attack success rate (ASR) across all possible target labels. Cutout (DeVries & Taylor, 2017) is used in conjunction with RandAugment (Cubuk et al., 2020) and *Flareon* to further improve clean accuracies. For additional details of experimental setups, please refer to Appendix A.

### 4.2 Flareon-Controlled Components

As *Flareon* assumes control of the data augmentation pipeline, this section investigates how *Flareon*-controlled hyperparameters affects the trade-offs between pairs of clean accuracies (CAs) and attack success rates (ASRs). Both $\beta$ and $\rho$ provide mechanisms to balance the saliency of shortcuts in triggers and the useful features to learn. Figure 3 shows that the perturbations added by the motion-based triggers are well-tolerated by models with improved trade-offs between CA and ASR for larger perturbations (smaller $\beta$). In addition, as we lower the perturbation scale of constant triggers with increasing $\beta$, it would require a higher proportion of images in a mini-batch with trigger added.

Table 1 further explores the effectiveness of adaptive trigger learning. As constant triggers with smaller perturbations (larger $\beta$) show greater impact on ASR, it is desirably to reduce the test-time perturbations added by them. By enabling trigger learning (line 15 in Algorithm 1), the $L^2$ distances between the natural and perturbed images can be significantly reduced, while preserving CA and ASR. Finally, Figure 4 visualizes the added perturbations.

Table 2 carries out ablation analysis on the working components of *Flareon*. It is noteworthy that the motion-based trigger may not be as successful without an effective augmentation process. Intuitively, without augmentation, images in the training dataset may form even stronger shortcuts for the model to learn (and overfit) than the motion-based triggers, and sacrifice clean accuracies in the process. Additionally, replacing the motion-based transform with uniformly-sampled pixel-wise

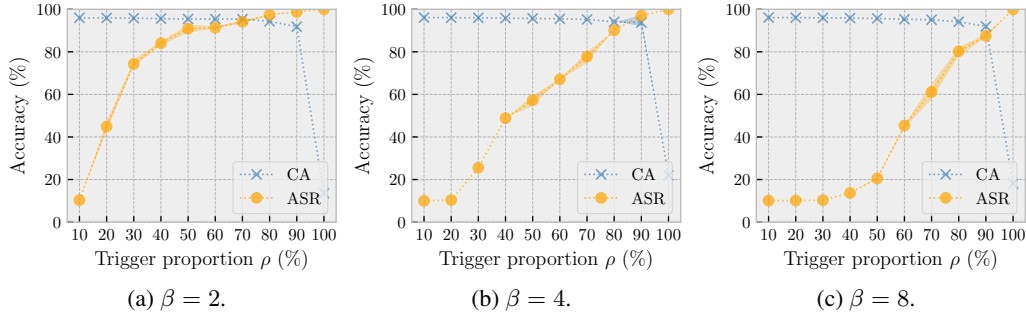

Figure 3: Effect of varying trigger initialization $\beta \in \{2, 4, 8\}$ and $\rho \in [10\%, 100\%]$ for constant triggers. The trigger ratio $\rho$ provides a mechanism to tune the trade-off between CA and ASR, and lower $\beta$ improves ASR, but with increasing perturbation scales. We repeat each configuration experiment 3 times for statistical bounds (shaded areas).

Table 1: Comparing the noise added ($L^2$ distances from natural images) by constant and adaptive triggers and their respective clean accuracies (%) and attack success rates (%).

| **CIFAR-10** | Constant trigger, $\beta =$ | | | | Learned trigger, $\epsilon =$ | | |
|---|---|---|---|---|---|---|---|
| Hyperparameters | 1 | 2 | 4 | 8 | 0.3 | 0.2 | 0.1 |
| $L^2$ distance | 1.99 | 1.65 | 1.27 | 0.92 | 0.88 | 0.67 | 0.39 |
| Clean accuracy (%) | 94.49 | 94.43 | 94.11 | 94.10 | 95.34 | 95.15 | 95.10 |
| Attack success rate (%) | 98.82 | 97.88 | 90.08 | 82.51 | 94.31 | 91.76 | 84.23 |

| **Datasets** | CelebA | | | | | *tiny*-ImageNet | | |
|---|---|---|---|---|---|---|---|---|
| Hyperparameters | | $\beta =$ | | | $\epsilon =$ | $\beta =$ | | $\epsilon =$ |
| | 1 | 2 | 4 | 8 | 0.01 | 1 | 2 | 0.2 |
| $L^2$ distance | 3.16 | 2.63 | 1.96 | 1.42 | 0.11 | 6.35 | 4.53 | 1.40 |
| Clean accuracy (%) | 78.88 | 80.11 | 79.87 | 79.69 | 78.20 | 57.14 | 57.23 | 55.42 |
| Attack success rate (%) | 99.98 | 99.88 | 99.16 | 99.89 | 99.40 | 98.44 | 74.23 | 79.14 |

triggers under the same $L^2$ distortion budget notably harms the resulting model's clean accuracy, adds visually perceptible noises, and can easily be detected with Grad-CAM (as shown in Figure 10 in the appendix).

### 4.3 TRAINER-CONTROLLED ENVIRONMENTS

The design of *Flareon* do not assume any prior knowledge on the model architecture and training hyperparameters, making it a versatile attack on a wide variety of training environments. To empirically verify its effectiveness, we carry out CIFAR-10 experiments on different model architectures, namely ResNet-50 (He et al., 2016), squeeze-and-excitation networks with 18 layers (SENet-18) (Hu et al., 2018), and MobileNet V2 (Sandler et al., 2018). Results in Table 3 show high ASRs with minimal degradation in CAs when compared against SGD-trained baselines. Table 4 presents additional results for CelebA and *tiny*-ImageNet that shows *Flareon* is effective across datasets and transform proportions $\rho$. Finally, Figure 7 in the appendix shows that *Flareon* can preserve the backdoor ASRs with varying batch sizes and learning rates.

### 4.4 DEFENSE EXPERIMENTS

As *Flareon* conceals itself within the data augmentation pipeline, it presents a challenge for train-time inspection to detect. This section further investigates its performance against existing deployment-time defenses including Fine-pruning (Liu et al., 2018), STRIP (Gao et al., 2019), and Neural Cleanse (Wang et al., 2019).

*Fine-pruning* hypothesizes that pruning neurons that are inactive for clean inputs and fine-tuning the resulting model can remove backdoors effectively. We test fine-pruning on the *Flareon*-backdoored models, and find backdoor neurons persist well against fine-pruning, as CAs can degrade at a faster

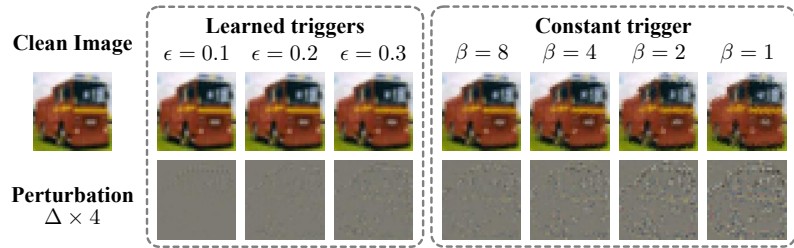

Figure 4: Visualizations of test-time perturbation noises (amplified $4\times$ for clarity) on CIFAR-10. Note that with larger $\beta$ values, the motion-based noise added to the original image becomes increasingly visible, whereas learned variants can notably reduce noise introduced by the trigger, while preserving high ASRs. For numerical comparisons, refer to Table 1.

Table 2: Ablation analysis of *Flareon*.

| Ablation of components | CA (%) | ASR (%) |
|---|---|---|
| No Augment | 92.26 | — |
| RandAugment (Cubuk et al., 2020) | 96.14 | — |
| AutoAugment (Cubuk et al., 2019) | 96.05 | — |
| *Flareon* with RandAugment and $\beta = 2$ | 95.35 | 94.12 |
| *Flareon* with AutoAugment and $\beta = 2$ | 95.16 | 97.01 |
| *Flareon* with no augment and $\beta = 2$ | 78.23 | 65.91 |
| *Flareon* with pixel-wise triggers ($\mathcal{T}_{\tau}(\mathbf{x}, y) = \mathbf{x} + \tau_y$) | 88.27 | 99.42 |

rate than ASRs *w.r.t.* channel sparsity (Figure 5a). *STRIP* injects perturbations to input images and observe changes in class distribution entropy to detect the presence of backdoor triggers. Figure 5 shows that the entropy distribution of *Flareon* models is similar to that of the clean model. *Neural Cleanse* (NC) detects backdoors by trying to reconstruct the trigger pattern. Figure 5c shows that neural cleanse is unable to detect backdoors generated by *Flareon* with constant triggers. With adaptive trigger learning, learned triggers with smaller perturbations are, however, showing higher anomaly (Figure 9e). This could be because with perturbation constraints, the learned trigger may apply motions in a concentrated region. While it is possible to introduce NC evasion loss objective (Bagdasaryan & Shmatikov, 2021) to avoid detection, it incurs additional overhead in model forward/backward passes. To defend against NC with *Flareon*, it is thus best to adopt randomly initialized constant triggers.

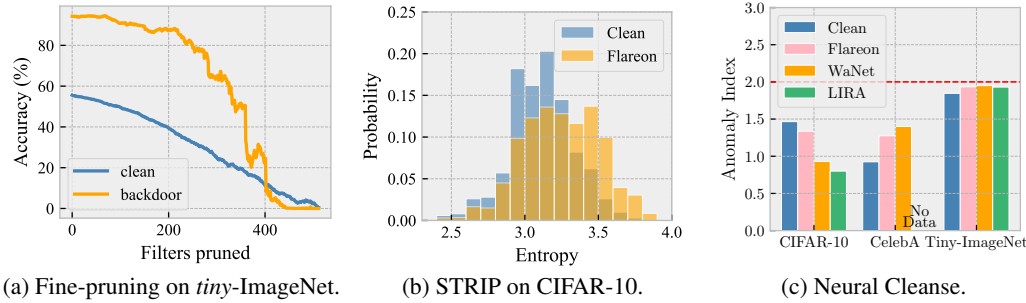

(a) Fine-pruning on *tiny*-ImageNet.  (b) STRIP on CIFAR-10.  (c) Neural Cleanse.

Figure 5: (a) Fine-pruning for the *tiny*-ImageNet model. (b) STRIP defenses on *Flareon* models. (c) comparing NC defenses against WaNet (Nguyen & Tran, 2020) and LIRA (Doan et al., 2021).

## 4.5 ADDITIONAL RESULTS

Table 5 compares recent SOTA backdoor attacks from the perspective of code-injection practicality. Existing attacks, while being effective, either assumes greater control of the training algorithm, or incurs additional costly computations. They additionally restrict attack possibilities on the trained

Table 3: Robustness against architecture choices.

| Architecture | Baseline (%) | CA (%) | ASR (%) |
|---|---|---|---|
| ResNet-50 (He et al., 2016) | 96.04 | 95.83 | 94.15 |
| SENet-18 (Hu et al., 2018) | 95.37 | 95.12 | 94.35 |
| MobileNet V2 (Sandler et al., 2018) | 95.34 | 94.59 | 97.28 |

Table 4: Robustness against dataset choices. CA and ASR values are all percentages. Varying test-time stealthiness $\beta$ and transform proportion $\rho$ for constant triggers. Rows with $\rho = 0\%$ show the baseline CAs without performing attacks.

| Datasets | $\rho$ (%) | $\beta = 1$ CA | $\beta = 1$ ASR | $\beta = 2$ CA | $\beta = 2$ ASR |
|---|---|---|---|---|---|
| CelebA | 0 | 78.92 | | | |
| | 70 | 79.13 | 99.85 | 78.87 | 99.41 |
| | 80 | 78.88 | 99.98 | 80.11 | 99.88 |
| *tiny*-ImageNet | 0 | 58.84 | | | |
| | 70 | 57.85 | 94.72 | 57.76 | 43.75 |
| | 80 | 57.14 | 98.44 | 57.23 | 74.23 |
| | 85 | 55.36 | 99.72 | 56.99 | 94.27 |
| | 90 | 54.05 | 99.72 | 55.06 | 96.57 |

Table 5: Comparing the assumed capabilities of SOTA backdoor attacks. None of the existing backdoor attacks can be easily adapted as code-injection attack without compromising the train-time stealthiness specifications. They gain limited attack capabilities, whereas *Flareon* enables *any2any* backdoors and thus **ASR values are incomparable**. "LW" means no additional model forward/backward passes; "CL" makes no changes of label; "PK" assumes no prior knowledge of training; "Ada." denotes learned triggers; and "St." indicates train-time and test-time stealthiness of trigger, ○ denotes partial fulfillment. "Target $\pi(y)$" represents possible test-time attack target transformations, here $y$ is the ground-truth label of the image under attack, and $s$ and $t$ are constant labels. We reproduce values with official implementation with default hyperparameters, except: "⋆" indicate data from the original literature, and "○" values are from BackdoorBench (Wu et al., 2022). Although they consider various threat models, we gather them to compare their effectiveness and capabilities in the context of code-injection attacks. [†]LIRA official results have no decimal precision. [‡]NARCISSUS uses a larger model than our ResNet-18 on *tiny*-ImageNet.

| Method | LW | CL | PK | Ada. | St. | Target $\pi(y)$ | CIFAR-10 CA | CIFAR-10 ASR of $\pi(y)$ | *tiny*-ImageNet CA | *tiny*-ImageNet ASR of $\pi(y)$ |
|---|---|---|---|---|---|---|---|---|---|---|
| WaNet (Nguyen & Tran, 2020) | ✓ | | | | ○ | $y \to t$ | 95.06 | 99.24 | 57.05 | 86.98 |
| LIRA (Doan et al., 2021)[†] | | | | ✓ | ✓ | $y \to y+1$ | 70.24 | 100.00 | 58. ⋆ | 59. ⋆ |
| Sleeper Agent (Souri et al., 2022) | | ✓ | ✓ | | ○ | $s \to t$ | 90.16 | 77.44 | 56.92° | 6.00° |
| Label Consistent (Turner et al., 2019) | | ✓ | | | | $y \to t$ | 89.30 | 98.47 | 57.03° | 9.84° |
| NARCISSUS (Zeng et al., 2022)[‡] | ✓ | ✓ | | ✓ | ○ | $y \to t$ | 95.07 | 98.44 | 64.65⋆ | 85.81⋆ |
| *Flareon* | ✓ | ✓ | ✓ | ✓ | ✓ | *any2any* | 95.21 | 98.81 | 56.99 | 94.27 |

model, typically requiring a pre-specified target, or label-target mapping. Finally, additional empirical results are in Appendix C, which includes more defense experiments.

## 5 CONCLUSION

This work presents *Flareon*, a simple, stealthy, mostly-free, and yet effective backdoor attack that specifically targets the data augmentation pipeline. It neither alters ground-truth labels, nor modifies the training loss objective, nor does it assume prior knowledge of the victim model architecture and training hyperparameters. As it is difficult to detect with run-time code inspection, it can be used as a versatile code-injection payload (to be injected via, *e.g.*, dependency confusion, name-squatting, or feature proposals) that disguises itself as a powerful data augmentation pipeline. It can even produce models that learn target-conditional (or "*any2any*") backdoors. Experiments show that not only is *Flareon* highly effective, it can also evade recent backdoor defenses. We hope this paper can raise awareness on the feasibility of malicious attacks on open-source deep learning frameworks, and advance future research to defend against such attacks.

## 6 REPRODUCIBILITY STATEMENT

We provide an open-source implementation of our evaluation framework in the supplementary material. All experiments in the paper uses public datasets, *e.g.*, CIFAR-10, CelebA, *tiny*-ImageNet. Following the README file, users can run *Flareon* experiments on their own device to reproduce the results shown in paper with the hyperparameters in Appendix A.

## 7 ETHICS STATEMENT

We are aware that the method proposed in this paper may have the potential to be used by a malicious party. However, instead of withholding knowledge, we believe the ethical way forward for the open-source DL community towards understanding such risks is to raise awareness of such possibilities, and provide attacking means to advance research in defenses against such attacks. Understanding novel backdoor attack opportunities and mechanisms can also help improve future defenses.

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

## A EXPERIMENTAL SETUP

### A.1 DATASETS

**CIFAR-10** consists of 60,000 $32 \times 32$ resolution images, of which 50,000 images are the training set and 10,000 are the test set. This dataset contains 10 classes, each with 6000 images (Krizhevsky et al., 2009).

**CelebA** is a large face dataset containing 10,177 identities with 202,599 face images. Following previous work (Saha et al., 2020), we select three balanced attributes from the 40 attributes: heavy makeup, mouth slightly, and smile, and combine the three attributes into 8 classes. For training, the baseline uses no augmentations on the images.

**Tiny-ImageNet** is an image classification dataset containing 200 categories, each category with 500 training images, 50 validation and 50 test images (Le & Yang, 2015). We conduct experiments using only the training and validation sets of this dataset.

Table 6 shows the details of these datasets.

Table 6: Overview of the datasets used in this paper.

| Dataset | | Input size | Train-set | Test-set | Classes |
|---|---|---|---|---|---|
| CIFAR-10 | ‖ | $32 \times 32 \times 3$ | 50,000 | 10,000 | 10 |
| CelebA | ‖ | $64 \times 64 \times 3$ | 162,770 | 19,962 | 8 |
| *tiny*-ImageNet | ‖ | $64 \times 64 \times 3$ | 100,000 | 10,000 | 200 |

### A.2 MODELS AND HYPERPARAMETERS

We evaluate *Flareon* using ResNet-18, MobileNet-v2, and SENet-18. The optimizer for all experiments uses SGD with a momentum of 0.9. Tables 7 and 8 provides the default hyperparameters used to train *Flareon* models.

Table 7: Default hyperparameters for constant *Flareon* triggers.

| Dataset | | CIFAR-10 | CelebA | *tiny*-ImageNet |
|---|---|---|---|---|
| Model learning rate $\alpha_{\text{model}}$ | ‖ | 0.01 | 0.01 | 0.01 |
| Model learning rate decay | ‖ | 1/2 every 30 epochs | None | 1/2 every 30 epochs |
| Weight decay | ‖ | 5e-4 | 5e-4 | 5e-4 |
| Epochs | ‖ | 350 | 50 | 400 |
| Batch size | ‖ | 128 | 128 | 128 |

Table 8: Default hyperparameters for adaptive *Flareon* triggers.

| Dataset | | CIFAR-10 | CelebA | *tiny*-ImageNet |
|---|---|---|---|---|
| Model learning rate $\alpha_{\text{model}}$ | ‖ | 0.01 | 0.01 | 0.01 |
| Model learning rate decay | ‖ | 1/2 every 30 epochs | None | 1/2 every 30 epochs |
| Trigger learning rate $\alpha_{\text{flareon}}$ | ‖ | 0.2 | 0.2 | 0.2 |
| Weight decay | ‖ | 5e-4 | 5e-4 | 5e-4 |
| Epochs | ‖ | 400 | 80 | 600 |
| Batch size | ‖ | 128 | 128 | 128 |

## B TRIGGER VISUALIZATIONS

In this section, we show the visualization of triggers on CelebA and *tiny*-ImageNet. Figure 6 show the clean samples and the samples after applying the motion-based triggers.

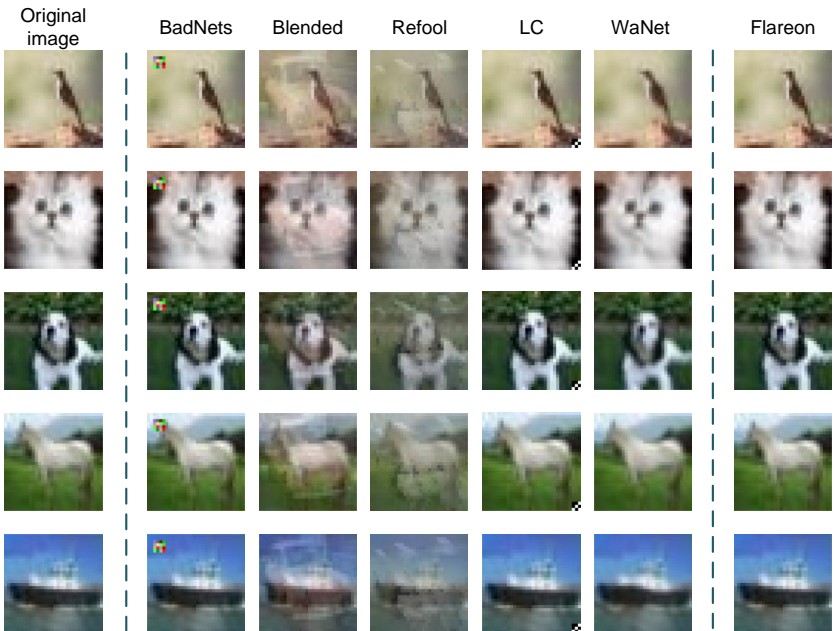

Figure 6: Comparing the test-time triggers of recent backdoor attacks (Patched (Gu et al., 2017), Blended (Chen et al., 2017), Refool (Liu et al., 2020), LC (Turner et al., 2019), and WaNet (Nguyen & Tran, 2020)).

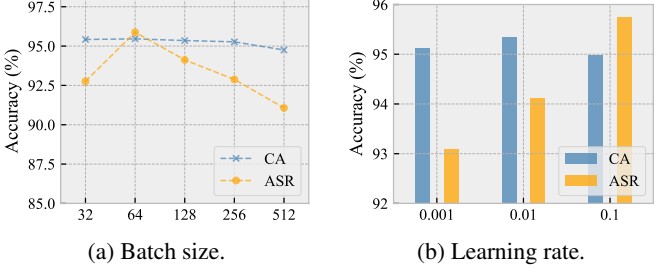

(a) Batch size.          (b) Learning rate.

Figure 7: Varying batch sizes and learning rates.

## C   ADDITIONAL RESULTS

Figure 7 shows that *Flareon* can preserve the backdoor ASRs with varying batch sizes and learning rates. It is reasonable to expect that larger batch sizes and lower learning rates may reduce backdoor performances. Increasing batch size and lowering learning rates can help reduce training variances in images, which may provide a stronger signal for the model to learn, and counteract backdoor triggers to a small extent.

We additionally compare the use of Uniform $\mathcal{U}(-s, s)$, Beta $\mathcal{B}(\beta, \beta)$, and Gaussian $\mathcal{N}(0, \sigma)$ initialized triggers in Table 9. Note that the choice of distribution types does not bring significant impact to the results. The rationale of choosing a Beta distribution is because it is nicely bounded within $[-1, 1]$, effectively limiting the perturbation of each pixel to be within its immediate neighbors. Besides, Beta distributions encompass Uniform distribution, *i.e.*, $\mathcal{B}(\beta, \beta)$ is Uniform when $\beta = 1$. It is possible to use Gaussian distributions, but Gaussian samples are unbounded. Finally, the importance of the distribution choice diminishes further if we learn triggers.

We visualize the confusion matrix and ASR matrix of the *Flareon*-trained CIFAR-10 model. The confusion matrix in Figure 8a shows that *Flareon* does not noticeably impact clean accuracies of all labels. Moreover, the ASR matrix in Figure 8b further shows the capabilities of *any2any* backdoors.

Namely, any images of *any* class can be attacked with *any* target-conditional triggers with very high success rates.

Table 9: Ablation on different distribution choices (Uniform $\mathcal{U}(-s, s)$, Beta $\mathcal{B}(\beta, \beta)$, and Gaussian $\mathcal{N}(0, \sigma)$) on the trigger initialization of *Flareon* on CIFAR-10, sorted by $L^2$ distances in ascending order. Note that Beta $\mathcal{B}(1, 1)$ is equivalent to the Uniform sampling within $[-1, 1]$. Beta distribution with $\beta = 2$ has better ASR with lower $L^2$ changes. The importance of initialization diminishes if we learn triggers. We rerun each setting 5 times with different seeds for statistical bounds.

| Distribution | $L^2$ distance ($\downarrow$) | Clean accuracy (%) | Attack success rate (%) |
|---|---|---|---|
| Uniform ($s = 0.70$) | $1.50 \pm 0.05$ | $94.51 \pm 0.32$ | $92.66 \pm 0.52$ |
| Uniform ($s = 0.75$) | $1.61 \pm 0.07$ | $94.22 \pm 0.12$ | $93.74 \pm 0.66$ |
| Beta ($\beta = 2$) | $1.67 \pm 0.07$ | $94.29 \pm 0.14$ | $97.25 \pm 0.63$ |
| Uniform ($s = 0.8$) | $1.77 \pm 0.09$ | $94.21 \pm 0.22$ | $95.51 \pm 1.04$ |
| Gaussian ($\sigma = 0.5$) | $1.84 \pm 0.06$ | $94.73 \pm 0.09$ | $91.24 \pm 2.13$ |
| Beta ($\beta = 1$) | $2.04 \pm 0.12$ | $94.41 \pm 0.08$ | $98.80 \pm 0.07$ |
| Gaussian ($\sigma = 0.75$) | $2.74 \pm 0.11$ | $94.13 \pm 0.14$ | $95.17 \pm 0.76$ |

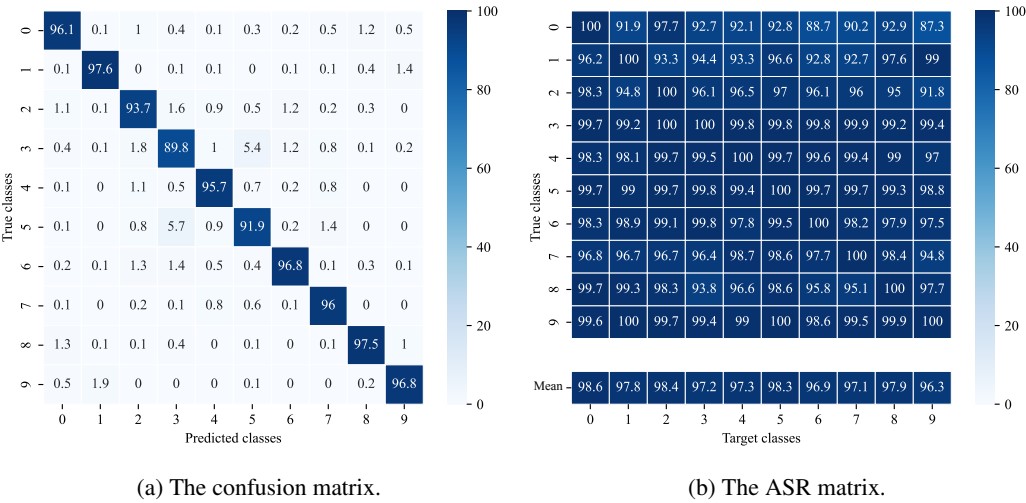

(a) The confusion matrix.  (b) The ASR matrix.

Figure 8: Class-wise statistics for the CIFAR-10 model. (a) The confusion matrix between the model prediction and ground-truth classes. (b) The ASR matrix shows the ASR values of attacking all test images of any label with any target class. "Mean" reports the overall ASR of each target.

## C.1 Defense Experiments

Figure 9 provides additional defense results. Visualization tools such as Grad-CAM (Selvaraju et al., 2017) are helpful in providing visual explanations of neural networks. Following Nguyen & Tran (2020), we also evaluate the behavior of backdoored models against such tools. Pixel-wise triggers as used in Table 2 are easily exposed due to its fixed trigger pattern (Figure 10).

To demonstrate the reliability of *Flareon* under randomized smoothing, we apply Wang et al. (2020) on *Flareon* with different trigger proportions $\rho$, as shown in the Table 10. In addition, we follow the setup of RAB (Weber et al., 2020), an ensemble-based randomized smoothing defense, and use the official implementation for empirical robustness evaluation, which sets the number of sampled noise vectors to $N = 1000$, and samples the smoothing noise from the Gaussian distribution $\mathcal{N}(0, 0.2)$ on CIFAR-10. For fairness, we use the same CNN model and evaluation methodology in RAB. The experimental results are in Table 11. *Flareon* enjoys great success under smoothing-based defenses.

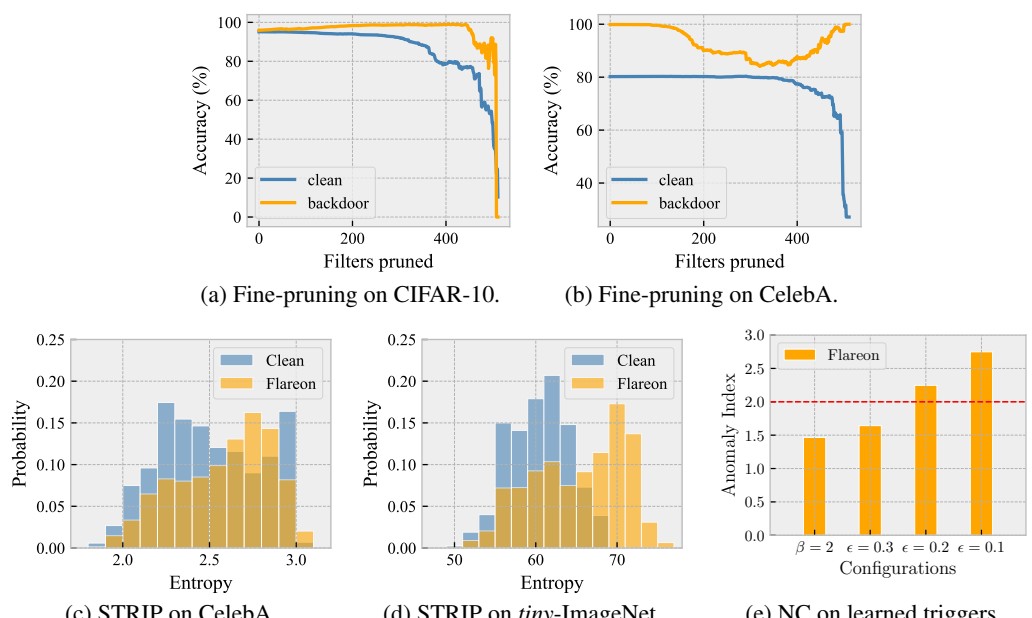

(a) Fine-pruning on CIFAR-10.  (b) Fine-pruning on CelebA.

(c) STRIP on CelebA.  (d) STRIP on *tiny*-ImageNet.  (e) NC on learned triggers.

Figure 9: (a, b) Fine-pruning for the CIFAR-10 and CelebA models. (c, d) STRIP defenses on CelebA and *tiny*-ImageNet models. (e) Smaller perturbations are easier to detect for neural cleanse.

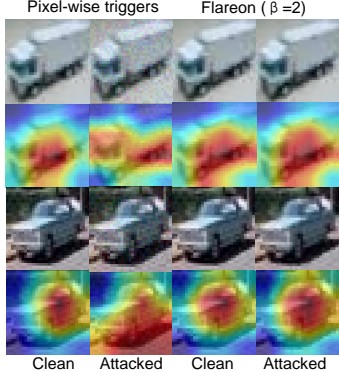

Figure 10: Grad-CAM heat maps and perturbed images comparisons between *Flareon* and pixel-wise triggers.

Table 10: Evaluation of randomized smoothing on *Flareon*.

| Model | $\rho =$ | 50 | 60 | 70 | 80 | 90 |
|---|---|---|---|---|---|---|
| CIFAR-10 | Clean accuracy (%) | 92.24 | 87.82 | 85.38 | 76.37 | 63.72 |
| | Attack success rate (%) | 97.33 | 96.70 | 98.10 | 99.42 | 99.16 |

Table 11: Evaluation of RAB on *Flareon*. "Vanilla" denotes training without RAB. Following Weber et al. (2020) for evaluation, the empirical robust accuracy reports the proportion of malicious inputs that not only attacks the vanilla model successfully, but also tricks RAB.

| Model | Benign Accuracy (%) | | Empirical Robust Accuracy under *Flareon* (%) | | | | |
|---|---|---|---|---|---|---|---|
| | Vanilla | RAB | Vanilla | $\rho = 50\%$ | $\rho = 60\%$ | $\rho = 70\%$ | $\rho = 80\%$ |
| CIFAR-10 | 61.71 | 58.74 | 0 | 9.71 | 8.15 | 6.45 | 3.82 |

## C.2 DISCUSSION AND RESULTS OF TURNER ET AL. (2019)

Label-consistent backdoor attacks (LC) Turner et al. (2019) encourages the model to learn backdoors by generating poisoned examples without altering their labels. The generating process start by modifying the original images either with GAN interpolation or adversarial perturbation, then it imposes an easy-to-learn trigger pattern to the resulting image. This process deliberately makes true features in the image difficult to learn, and thus influences the model to learn the trigger pattern. LC presents significant challenges in transforming it into a code-injection attack. The reasons are as follows:

1. The triggers are clearly visible to human (Figure 6).
2. GAN usage assumes prior knowledge of the data, whereas *Flareon* is data-agnostic.
3. Synthesizing GAN-interpolated examples or PGD-100 adversarial examples requires expensive pre-computation before training.
4. Even if they are directly deployed as code-injection attacks, run-time profiling inspections, *e.g.*, with PyTorch profiler will reveal both approaches contain erroneous unwanted computations. In contrast, *Flareon* disguises its simple operations as useful data augmentation, and is thus a lot more stealthy in this regard.
5. Because of the constant triggers and harmful alterations to the original images, We show that LC is unlikely to be effective against NC (Figure 11), and they are also impactful on clean accuracies (Table 12).

Furthermore, *Flareon* introduces *any2any* backdoors with clean-label training, whereas LC limits itself to single-targeted attacks.

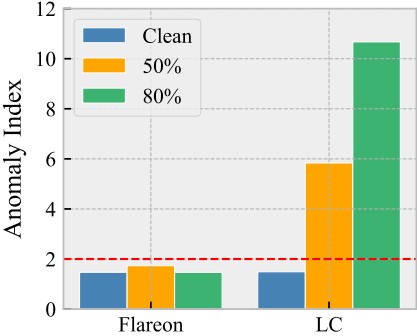

Figure 11: Comparing NC on *Flareon* and LC under different trigger proportions $\rho \in \{50\%, 80\%\}$. Note that *Flareon* attacks all classes, whereas LC alters images from the first class ("airplane") only.

Table 12: Comparing LC (Turner et al., 2019) with PGD-100 and *Flareon* on CIFAR-10 in terms of clean accuracies. We remind that $\beta$ is used for trigger initialization and larger values indicate stealthier triggers. Here, $y \rightarrow t$ mean single-targeted attack. To compare with LC, we provide results that restrict *Flareon*'s capability to single-target poisoning only, which translates to poisoning $\rho/10$ of all training examples per mini-batch on CIFAR-10.

| | LC $y \rightarrow t$, PGD-100 | | *Flareon* $y \rightarrow t$, $\beta = 1$ | | *any2any*, $\beta = 2$ | |
|---|---|---|---|---|---|---|
| Baseline accuracy without attack (%) | 92.53 | | 96.14 | | 96.14 | |
| Average poisoned samples per batch | 5% | 8% | 5% | 8% | 50% | 80% |
| Clean accuracy (%) | 89.61 | 89.30 | **95.70** | 94.41 | 94.22 | 94.43 |
| $\Delta$ Clean accuracy (%) | $-2.92$ | $-3.23$ | $\mathbf{-0.44}$ | $-1.73$ | $-1.92$ | $-1.71$ |
| Attack success rate (%) | 81.47 | 96.00 | 85.32 | 98.60 | 93.14 | 97.78 |

