# OpenReview forum: "Flareon: Stealthy Backdoor Injection via Poisoned Augmentation"
_ICLR.cc/2023/Conference — Submitted to ICLR 2023_

### Official Review · Reviewer_J6kD · 2022-10-13

**Confidence:** 4
**Correctness:** 2
**Technical Novelty And Significance:** 2
**Empirical Novelty And Significance:** 2
**Recommendation:** 3

**Clarity, Quality, Novelty And Reproducibility:**

The paper lacks technical clarity and novelty (See above for details).
Since the implementation has been released, I believe the results are reproducible.

**Details Of Ethics Concerns:**

I do not have ethical concerns about this work.

**Strength And Weaknesses:**

Strengths
+ The paper conducts a reasonably large set of experiments, and the results are promising.
+ The authors open-source the implementation of the proposed technique.

Weaknesses:
1. My first concern about this work is its novelty and technical contribution. The idea of the proposed method is similar with an existing technique, which is also cited in the paper, i.e., Turner et al 2019. In that paper, the authors proposed clean-label attacks (label-consistent attack). This set of attacks also adds triggers to the samples from the target class and does not vary the label of the original samples. To do so, it also proposed to break the semantic features in an original samples, add a trigger, and force the model to learn the shortcuts from the trigger to the target class. To achieve this, it proposes two methods, one is GAN-based generation, the other is adversarial perturbation-based method. The latter is similar with the method in this work. Especially, Line 15 in Algorithm1 of this work is actually generating adversarial examples. As such, I am afraid that this work has limited technical novelty and contribution. Besides, the authors' argument about the limitation of Turner et al 2019 is also not convincing enough. It states Turner et al 2019 introduce too much additional computational cost and is not applicable to the problem of this work. As I discussed above, the perturbation-based method in Turner et al 2019 is similar to the one proposed by this work, which also require only gradient computation. If the method proposed in this work is acceptable, so should be the one proposed in Turner et al 2019. Besides, regarding the GAN-based method, if GAN is pretrained, the computational cost of generating adversarial trigger is also marginal. Therefore, I am not fully convinced by the argument. Overall, I belive the paper does not well distinguish itself from an existing technique and thus lack novelty.

2. The proposed technique is not clearly introduced. Some important details are missing. I belive the proposed motion-based trigger generation deserve a more detailed description since it is one of the main technical novelties. The current description is not entire clear. Frist, I would suggest the authors explain what motion-based stands for. Second, I would suggest explaining the design intuition of each step, for example, why using beta distribution. Can we use gaussion or uniform instead? Third, the paper does not explain what "grid_sample" stands for. Finally, visually, the triggers in Figure 4 are similar to pixel-level adversarial perturbation. I do not fully understand the difference between these two. Actually, Line 15 is generating adversarial perturbation. Can we just use Line 15 without the proposed trigger transformation?

3. The evaluation on defense misses: (1) An important baseline (Turner et al 2019); (2) One type of defense works that learns a robust model from poisoned samples (e.g., [Rab: provable robustness against backdoor attacks]).


**Summary Of The Paper:**

This paper proposes a new backdoor attack that does not alter the label of the poisoned samples, a.k.a, clean-label attack. The high-level idea is to craft a trigger such that it breaks the semantic features in an input image and adds a trigger that is easier to be learned. Adding this trigger to the input and training the model could force a learning model to pick up the association between the backdoor trigger and the sample label rather than the semantic features between the sample label. The paper evaluates its attack on three datasets and against three defenses to demonstrate its exploitability.

**Summary Of The Review:**

This paper proposes a new backdoor attack against DNNs. As I mentioned above, the proposed technique is similar to the existing clean-label attack and lacks technical novelty. The evaluation is not comprehensive, missing important baselines and possible defenses. As such, I believe this work may not be able to meet the bar of ICLR, a top-tier conference in ML.

---

> ### Author Response · Authors · 2022-11-07
> **1: Explaining the core contributions/novelty of this paper.**
>
> Thank you for reviewing our paper and providing feedback on our work. We would like to address your concerns below, and we are revising our submission for the suggested changes.
>
> To begin, we would like to emphasize that the core contribution/novelty of this paper is the following:
>
> * It demonstrates the feasibility of **stealthy code-injection payload** that can potentially have great ramifications on open-source frameworks; and
> * It **introduces any2any backdoors** in backdoored models with clean-label training.
>
> Both findings above are not present in existing work, including label-consistent backdoor attacks (LC) [Turner et al., 2019]. It was **previously unknown** that such a **small, seemingly innocuous change** of the augmentation pipeline could have a **large ramification on the trained model**, and Flareon makes this surprising discovery.

---

> ### Author Response · Authors · 2022-11-07
> **2: Addressing each point of concern.**
>
> ### Regarding the novelty of Flareon:
>
> > In [Turner et al 2019] the adversarial perturbation-based method is similar with the method in this work.
>
> We would like to draw important distinctions between Flareon and Label-consistent backdoor attacks (LC) [Turner et al., 2019]:
>
> 1. LC did not discover the existence of **any2any backdoors**.
> 2. LC with PGD incurs substantial overhead, and LC with GAN further requires prior knowledge for GANs. Both present significant challenges in applying to our novel code-injection threat model.
> 3. It uses **constant patch-based** triggers, which are not stealthy, whereas Flareon uses motion-based triggers that generates instance-specific pixel-wise perturbations and further enables **learning of triggers**.
>
> > Especially, Line 15 in Algorithm1 of this work is actually generating adversarial examples.
>
> **This is not true.** We kindly point out that Line 15 of Algorithm 1 is performing **gradient descent** on the trigger parameters (note the minus sign in $\tau -\alpha_{\mathrm{flareon}}\nabla_{\tau} \ell$. This is the **exact opposite** of generating adversarial examples. Rather, it minimizes the SCE loss by optimizing the trigger.
>
> In addition, eq. (3) in Section 3.1 clarifies our objective to be a *joint minimization* of the classification loss w.r.t. the model parameters and triggers.
>
> > [Turner et al 2019] requires only gradient computation. […] if GAN is pretrained, the computational cost of generating adversarial trigger is also marginal.
>
> 1. We kindly highlight that LC [Turner et al., 2019] does not generate triggers through either PGD or GAN. Rather, it imposes constant patch-based triggers onto images, see [Figure 2 and Section 4.4 of [Turner et al., 2019]](https://arxiv.org/pdf/1912.02771.pdf) for clarification. The triggers added by LC are thus not stealthy.
>
> 2. The first variant of LC uses PGD-100 to generate adversarial examples, and then add constant patch-based triggers to these adversarial images to generate poisoned samples. If used in our novel threat model, LC would thus require 100 forward/backward passes per poisoned image — a substantial overhead.
>
> 3. It is also impractical for the code-injection threat model to use GANs, as it would require prior knowledge of the training data distribution (for either pre-trained GANs or the training of them), even without consideration of computational overheads.
>
> We will include LC in both Table 5 and the comparisons of triggers in Figure 6 in the appendix to clarify this further.  **[$\checkmark$ Done.]**
>
> Finally, LC presents significant challenges in transforming it into a code-injection attack. The reasons are as follows:
> 1. The triggers are visible to human.
> 2. GAN training or pre-trained GANs assume prior knowledge of the data, whereas Flareon is data-agnostic.
> 3. Even if they are directly deployed as code-injection attacks, profiling inspections, e.g. with [PyTorch profiler](https://pytorch.org/tutorials/recipes/recipes/profiler_recipe.html) will reveal both contain erroneous unwanted computations, whereas Flareon disguises its simple operations as useful data augmentation, and is thus a lot more stealthy in this regard.
> 4. We hypothesize that LC approaches are unlikely to be effective against NC and other defenses, because of the constant triggers, and they are also impactful on clean accuracies. We are running experiments to confirm these hypotheses.

---

> ### Author Response · Authors · 2022-11-07
> **3: Addressing each point of concern. Continued.**
>
>  ### Regarding the technical details:
> > Frist, I would suggest the authors explain what motion-based stands for.
>
> It means that instead of adding pixel-wise perturbations, it computes output images using an input image, and a flow-field specifying how pixels are moved for each pixel.
>
> This is performed with `torch.nn.functional.grid_sample`, please refer to this [PyTorch documentation](https://pytorch.org/docs/stable/generated/torch.nn.functional.grid_sample.html).
>
> > Second, I would suggest explaining the design intuition of each step, for example, why using beta distribution. Can we use gaussion or uniform instead?
>
> The rationale of choosing a Beta distribution is because it is bounded within [-1, 1], effectively limiting the perturbation of each pixel to be within its immediate neighbors. Besides, Beta distributions encompass Uniform distribution, i.e., $\mathcal{B}(\beta, \beta)$ is Uniform when $\beta = 1$. It is entirely possible to use Gaussian and Uniform distributions, but Gaussian samples are unbounded, and Uniform does not have a tuning parameter for variance. We will report results with Gaussian and Uniform triggers and reply soon. **[$\checkmark$ Done. Please see Table 9 in Appendix C]**
>
> We kindly remind that the choices of distributional settings are irrelevant to our core contribution, and they can only serve as an initialization for the trained triggers.
>
> > Third, the paper does not explain what “grid_sample” stands for.
> > Finally, visually, the triggers in Figure 4 are similar to pixel-level adversarial perturbation. I do not fully understand the difference between these two.
>
> Please see the answer above for the first point. Figure 10 in the appendix further provides a comparison between the two types of triggers.
>
> > Actually, Line 15 is generating adversarial perturbation. Can we just use Line 15 without the proposed trigger transformation?
>
> As mentioned above, line 15 is not generating adversarial perturbation, but rather it is training a **loss-minimizing perturbation**. The last row of Table 2 replaces the motion-based trigger with a pixel-wise trigger, and shows that a trigger of similar $L^2$ perturbation would result in a large decrease in clean accuracy. Moreover, it results in visible changes in the Grad-CAM heat maps (Figure 10 in the appendix).
>
> ### Regarding the defenses:
>
> > (1) An important baseline (Turner et al 2019)
>
> We kindly note that [Turner et al., 2019] is not a defense method. We will include LC in both Table 5 and the comparisons of triggers in Figure 6 in the appendix to further differentiate our key contributions from their work. **[$\checkmark$ Done. We also added a discussion of LC in Appendix C.2.]**
>
> > (2) One type of defense works that learns a robust model from poisoned samples (e.g., [Rab: provable robustness against backdoor attacks]).
>
> We can certainly test smoothing-based defenses including RAB and will update you on this. We will provide our findings during the discussion period. **[$\checkmark$ Done. Please see Appendix C.1 Defense Experiments.]**

---

> ### Author Response · Authors · 2022-11-07
> **4: "Good design is as little design as possible. Less, but better."**
>
> The core contribution/novelty of this paper is not to improve existing backdoor attacks by considering their threat models. Rather, it investigates a **previously unexplored threat** to inject malicious and stealthy code in open-source DL frameworks. It presents **unique challenges that cannot be adequately addressed by existing backdoor attacks**, namely, the stringent requirements of train-time stealthiness (introduced in Section 1). Our approach further enables **any2any** attacks. Overall, the new discovery may greatly increase the practicability and value of backdoor attacks.
>
> The key takeaway is that such a **small, seemingly innocuous change** of the augmentation pipeline could have a **surprisingly large ramification on the trained model**, and we are glad that Flareon makes this discovery for the first time. We hope the reviewer can also share the excitement in this finding, and hope to raise awareness within the deep learning (DL) community of such an unexplored threat, in order to encourage research of future attacks and defenses on open-source DL frameworks, and to **better prepare us for and prevent such attacks from exacting heavy costs** on the industry.
>
> We again thank the reviewer for the feedback, and we hope to address your concerns adequately in the discussion period. We hope you can kindly offer our rebuttal a thoughtful response, and reconsider the current rating.
>
> Please let us know if you have any other comments/questions. In the meanwhile, we will further strengthen the manuscript by providing additional defense and ablation experiments.

---

> ### Author Response · Authors · 2022-12-05
> **A gentle reminder to discuss our paper**
>
> Dear reviewer,
>
> We hope we have addressed your concerns sufficiently in our rebuttal. We have **only one week** left in the discussion period. We still hope to hear back from you. Please let us know if there are remaining concerns, and reconsider the current rating.
>
> Thanks again for your review.
>
> Best,
>
> Paper 4225 authors.

---

### Official Review · Reviewer_t5Up · 2022-10-24

**Confidence:** 4
**Clarity, Quality, Novelty And Reproducibility:** See above for discussion of novelty.
**Correctness:** 4
**Technical Novelty And Significance:** 2
**Empirical Novelty And Significance:** 2
**Recommendation:** 3

**Strength And Weaknesses:**

Strengths: the presented method has consistently high adversarial success rates.

Weakness: the core weakness is novelty. The presented method is a straightforward extension from previous work. More details below.

Novelty - aspects of the presented algorithm that are new compared to previous work:
* Focus on data augmentation aspect of pipeline: Previous works can also be implemented as data augmentation (as they just add a class consistent perturbation i.e. [0, 1])
* Type of backdoor: [2] also uses a beta distribution to sample backdoor triggers
* Many types of triggers: straightforward extension, done previously in [1]
* Adversarial success rate: the presented method is consistently in the same ballpark as previous methods (see Table 5 in the paper)

[0] https://arxiv.org/abs/2002.00937 Section 3

[1] https://people.cs.uchicago.edu/~ravenben/publications/pdf/backdoor-sp19.pdf

[2] https://ojs.aaai.org/index.php/AAAI/article/view/17266

**Summary Of The Paper:**

In this work the authors present a data poisoning attack that inserts class-conditional triggers in the data augmentation stage.

The threat model the authors consider is:
* The defender is oblivious and uses the attacker's malicious data augmentation pipeline
* The attacker has arbitrary access to the data augmentation pipeline and wants to be able to use triggers at test-time that will make images classify as certain classes

The authors method is to insert class-consistent triggers (specifically: -1/1 valued vectors chosen according to a beta distribution for each target class). The method has consistently high adversarial success rates.

**Summary Of The Review:**

The presented method is not novel enough to warrant publication.

---

> ### Author Response · Authors · 2022-11-07
> **1: Explaining the core contributions/novelty of this paper.**
>
> Thank you for reviewing our paper and providing helpful feedback on our work. We would like to address your concerns related to novelty below.
>
> To begin, we would like to emphasize that the core contribution/novelty of this paper is the following:
>
> * It demonstrates the feasibility of **stealthy code-injection payload** that can potentially have great ramifications on open-source frameworks; and
> * It **introduces any2any backdoors** in backdoored models with clean-label training.
>
> Both claims above are not present in existing work. In Section 1 we discussed in depth the specific challenges in applying existing methods onto this threat model. It thus merits investigation and we are happy that Flareon provides a simple and highly effective solution. It was previously unknown that such a **small, seemingly innocuous change** of the augmentation pipeline could have **great ramifications on the trained models**, and Flareon **is the first to make this surprising discovery**. We hope the reviewer can also share the excitement of this finding.
>
> We kindly remind that the overall approach of this paper is *simple by design* to not only avoid diluting the inventive insight proposed in this paper, but to also satisfy the constraints posed by the novel threat model. Most importantly, we believe unnecessary complexity in design is not good science, as it does not add value to the core contribution. Increase in complexity also further reduces its general applicability.

---

> ### Author Response · Authors · 2022-11-07
> **2: Addressing each point of concern.**
>
> In addition, the related works cited by the reviewer can also not provide a satisfactory solution to this new challenge.
>
> > Focus on data augmentation aspect of pipeline: Previous works can also be implemented as data augmentation (as they just add a class consistent perturbation i.e. [0, 1])
>
> Radioactive data [0] does not consider the backdoor threat model, and cannot be used as a code-injection payload, and does not achieve backdoors. Neural cleanse [1] is a defense strategy, and does not propose class-consistent perturbations. It is thus unclear how [0] and [1] can be implemented as augmentation-based attacks.
>
> > Type of backdoor: [2] also uses a beta distribution to sample backdoor triggers
>
> The rationale of choosing a Beta distribution is because it is bounded within [-1, 1], effectively limiting the perturbation of each pixel to be within its immediate neighbors. DeHiB [2] does not overlap with our work as it considers data poisoning in semi-supervised learning. Beta distribution is a useful tool, and thus likely to be used frequently in unrelated DL works.
>
> We kindly remind that the choices of distributional settings are irrelevant to our core contribution, and they can only serve as initialization for the trained triggers. Please see Table 9 in Appendix C for a comparison between trigger initializations.
>
> > Many types of triggers: straightforward extension, done previously in [1]
>
> Neural Cleanse in [1] is a defense strategy, it considers many types of triggers that exist in related works. Flareon has two new types of triggers, i.e., its constant and learned versions, both of which are new and not in [1].
>
> > Adversarial success rate: the presented method is consistently in the same ballpark as previous methods (see Table 5 in the paper)
>
> We kindly remind the reviewer that all of the compared methods may seize greater freedom to attack, and none of which can be trivially adapted to the code-injection threat model without violating the attack specifications discussed in Section 1. For instance, they may require label flipping (e.g. LIRA and WaNet) or additional computational overhead (e.g. Hidden Trigger, Sleeper Agent). Furthermore, they only gain much more restrictive attack abilities (see the "Target" column) and Table 5 reports **their restrictive attack ASRs** from official publications (e.g., targeting 1 label only). In contrast, Flareon reports **any2any** ASR values over all targets in Table 5.
>
> TL;DR:  Flareon can gain more backdoor functionalities while being more restrained, so similar ASRs indicate a much greater practical value.

---

> ### Author Response · Authors · 2022-11-07
> **3: "Good design is as little design as possible. Less, but better."**
>
> We appreciate the reviewer’s effort to point out overlapping components from existing work, but we kindly remind that we did not claim those components as our main contributions, these are simply a means to the end. Rather, we intend to raise awareness within the deep learning (DL) community of **a simple, stealthy, seemingly harmless yet effective code-injection attack**, which also enables **highly versatile any2any backdoors** in trained models, in order to encourage research of future attacks and defenses on open-source DL frameworks, and to **better prepare us for and prevent such attacks from exacting heavy costs** on the industry.
>
> We again thank the reviewer for the feedback, and we hope to address your concerns adequately in the discussion period. We hope you can kindly offer our rebuttal a thoughtful response, and reconsider the current rating.
>
> Please let us know if you have any other comments/questions. In the meanwhile, we will further strengthen the manuscript by updating additional defense and ablation experiments.

---

> ### Author Response · Authors · 2022-12-05
> **A gentle reminder to discuss our paper.**
>
> Dear reviewer,
>
> We hope we have addressed your concerns sufficiently in our rebuttal. We have **only one week** left in the discussion period. We still hope to hear back from you. Please let us know if there are remaining concerns, and reconsider the current rating.
>
> Thanks again for your review.
>
> Best,
>
> Paper 4225 authors.

---

### Official Review · Reviewer_2ChX · 2022-10-24

**Confidence:** 4
**Correctness:** 3
**Technical Novelty And Significance:** 3
**Empirical Novelty And Significance:** 3
**Recommendation:** 6

**Clarity, Quality, Novelty And Reproducibility:**

Novelty:
- Flareon seems novel.

Quality:
- The statement that backdoor attacks assume full control of the training process is mostly not true. The majority of methods only assume full control of the dataset, without any changes to the training procedure.
- It is not clear to me how the proposed attack accomplishes the targeting of a specific label from a given clean label (any2any).
- Method limitations should be addressed in the paper:
  * Flareon seems limited to the image domain.
  * It seems that, for good performance of the proposed attack, ~80% of the images need to be backdoored. This is not particularly efficient.

Quality - experiments:
- The main comparison with other attacks (Tab. 5) seems to be using the numerical values from the original papers. Moreover, many values are missing, leading to an incomplete and skewed comparison. The baselines should preferably be re-run in the same conditions as the proposed method.
- The paper seems to cite other backdoor attacks that optimize the trigger (e.g., reference [Turner et al., 2019] in the paper), however these are not included as attack baselines.
- The literature review on backdoor and poisoning defenses is a bit light. Examples of additional references to include: [Wang et al., 2020], [Weber et al., 2020]. These could also be part of the experiments.
- It would be relevant to see how Flareon and the other attacks fare against defenses.

Clarity:
- Overall, a clear, well-structured paper.
- Additional proofreading would improve the quality of the paper.

Reproducibility:
- Open-source implementation provided.

Minor:
- Fig. 1 is a bit misleading, as it looks similar to an adversarial examples pipeline. It is not clear that the model is trained with the backdoor.
- tempering -> tampering
- code-inject -> code-injection
- and etc. -> etc.
- "motion-based triggers can successfully deceive recent backdoor attacks" -> "motion-based triggers can successfully deceive recent backdoor defenses"?

References
* [Wang et al., 2020] Binghui Wang, Xiaoyu Cao, Neil Zhenqiang Gong, et al. On certifying robustness against backdoor attacks via randomized smoothing. arXiv preprint arXiv:2002.11750, 2020.
* [Weber et al., 2020] Maurice Weber, Xiaojun Xu, Bojan Karlaš, Ce Zhang, and Bo Li. Rab: Provable robustness against backdoor attacks. arXiv preprint arXiv:2003.08904, 2020.


**Strength And Weaknesses:**

Strengths:
- Flareon seems to have many qualities, and solves some limitations of existing backdoor attacks.
- The topic of the paper is interesting and relevant to the ICLR community.
- Good experiments, investigating the effect of varying method parameters, but also evaluating the proposed method against state-of-the-art baselines.
- Well-written, clear paper.
- Open-source implementation.

Weaknesses:
- The experimental evaluation could be improved.
- The limitations of the proposed method are not discussed in the paper.

See more details below.

**Summary Of The Paper:**

This paper proposes Flareon, a new backdoor attack that is hidden in the data augmentation step commonly used when training computer vision models. The attack aims to remain stealthy: data labels are not changed, images only suffer small augmentations that add the triggers, and the memory and computation overhead of Flareon are relatively low. Moreover, this code injection attack is able to introduce backdoors that target any class starting from any original clean label. The method seems to reach a good clean accuracy-attack success rate trade-off on CIFAR-10, CelebA and Tiny ImageNet datasets.

**Summary Of The Review:**

Novel code injection attack that aims to overcome all limitations specific to backdoor attacks. Good evaluation, that could still be improved.

---

> ### Author Response · Authors · 2022-11-07
> **Thank you for reviewing our paper and providing helpful feedback on our work.**
>
> We would like to address your concerns below, and we are revising our submission for the suggested changes.
>
> > The statement that backdoor attacks assume full control of the training process is mostly not true. The majority of methods only assume full control of the dataset, without any changes to the training procedure.
>
> We agree with the reviewer that many of the backdoor attack methods do not assume full control of the training process. We only intended this claim to apply to BadNets, trojaning attacks, WaNet and LIRA. To clear up confusion, we have updated the text to reflect this.
>
> > It is not clear to me how the proposed attack accomplishes the targeting of a specific label from a given clean label (any2any).
>
> We assume you are asking about the test-time trigger application, and would like to answer your question as follows. For example, given any image, one can choose to apply the pre-trained motion-field trigger $\tau_\mathrm{Car}$ of label “car” onto the image, specifically with `torch.nn.fuctional.grid_sample` ([doc. here](https://pytorch.org/docs/stable/generated/torch.nn.functional.grid_sample.html)), by taking the image as input and the trigger $\tau_\mathrm{Car}$ as the flow-field. The backdoored model would incorrectly classify the resulting image as a “car” with high success rates, regardless of the ground-truth class of the input image.
>
> > Flareon does seem to be limited to the image domain.
>
> We will try to expand its application domain in the future, but we fall in line with existing backdoor attacks which also considers vision tasks only. The goal of this paper is to highlight the existence of a new code-injection threat model and provide practical attack payload solution for the DL community.
>
> > It seems that, for good performance of the proposed attack, ~80% of the images need to be backdoored. This is not particularly efficient.
>
> Perhaps could you clarify what you mean by “efficient” and we can provide a more specific answer to this question. As mentioned in Section 3.3, the added overhead by trigger application is minimal. We chose $\rho = 80\\%$ because it provides high attack success rates (ASRs) without simultaneously degrading clean accuracies (CAs). Under larger perturbations, $\rho$ can be reduced further.
>
> > The main comparison with other attacks (Tab. 5) seems to be using the numerical values from the original papers. Moreover, many values are missing, leading to an incomplete and skewed comparison.
>
> We are reproducing additional results in Table 5 with their respective official source codes, and will update you on this soon. **[$\checkmark$ Done.]** However, we kindly remind the reviewer that all of the compared methods may seize greater freedom to attack, and none of which can be trivially adapted to either the code-injection threat model without violating the attack specifications discussed in Section 1. Furthermore, they only gain much more restrictive attack abilities. Please note that our ASRs are *any2any* results, whereas the others are mostly single-target accuracies.
>
> > The baselines should preferably be re-run in the same conditions as the proposed method.
>
> Could you clarify what was meant by “the same conditions”? The related methods in Table 5 assume different threat models and different capabilities, and can thus not be used in our threat model without breaking subjective assumptions. They also present significant challenges to be adapted for train-time stealthiness. We are willing to reproduce them, but would like to receive clarification on this before carrying out experiments.
>
> > The paper seems to cite other backdoor attacks that optimize the trigger (e.g., reference [Turner et al., 2019] in the paper), however these are not included as attack baselines.
>
> We appreciate the reviewer’s suggestion for actionable experiments. We will add label-consistent backdoor attacks [Turner et al., 2019] for comparison in Table 5. **[$\checkmark$ Done. We also added a discussion of LC in Appendix C.2.]**
>
> > The literature review on backdoor and poisoning defenses is a bit light. Examples of additional references to include: [Wang et al., 2020], [Weber et al., 2020]. These could also be part of the experiments.
> > It would be relevant to see how Flareon and the other attacks fare against defenses.
>
> We can certainly test smoothing-based defenses [Wang et al., 2020], [Weber et al., 2020] and will update you on this. We will provide our findings during the discussion period.  **[$\checkmark$ Done. See Appendix C.1 Defense Experiments.]**
>
> > Minor: […]
>
> Thank you for pointing out the typos in the paper, and it has been updated accordingly.
>
> We again thank you for the detailed feedback, and we hope to address your concerns adequately in the discussion period to further strengthen the paper. Please let us know if you have any other comments/questions. In the meanwhile, we will further revise the manuscript by providing additional baseline, defense and ablation experiments.

---

> > ### Comment · Reviewer_2ChX · 2022-11-22
> > **Thank you for your response**
> >
> > Thank you for the additional details that have answered some of my questions, and for incorporating the feedback from the review. I also appreciate the new experimental results.
> >
> > Regarding your question:
> >
> > >>The baselines should preferably be re-run in the same conditions as the proposed method.
> > >
> > > Could you clarify what was meant by “the same conditions”?
> >
> > While we agree that the differences in threat model and access would not be addressed by re-running baselines, differences regarding other points in the setup are controlled when doing so (hardware, experimental protocol, etc.). Such gaps that can account for results variations can sometimes remain hidden when comparing directly to metrics from other papers.

---

> > > ### Author Response · Authors · 2022-11-28
> > > **Additional reproduced results as requested.**
> > >
> > > Thanks for your suggestion,
> > > and we note that we reproduced the CIFAR-10 results in Table 5,
> > > and we would like to offer the following results
> > > after re-running each of the methods for Tiny-ImageNet.
> > >
> > > Following the setup of the original papers,
> > > we reproduced the attacks with their official implementations,
> > > namely WaNet, LIRA, Sleeper Agent,
> > > label-consistent backdoor attacks (LC) and NARCISSUS.
> > > We note that all experiments evaluate ASRs
> > > by assuming their respective attack capabilities
> > > (specified by the "Target $\pi(y)$" column).
> > >
> > > The training hyper-parameters of both LIRA and Sleeper Agent
> > > are identical to the original publications.
> > > As LC did not use the Tiny-ImageNet dataset,
> > > we reproduced it using the experimental settings
> > > from BackdoorBench [1].
> > > We present our findings in the table below,
> > > where values "Nominal CA/ASR" columns
> > > are from the original papers,
> > > except values marked with "*" are from [1],
> > > as they were not reported in the original papers:
> > > | Tiny-ImageNet    | Target $\pi(y)$ | Nominal CA | Nominal ASR | Reproduced CA | Reproduced ASR |
> > > |------------------|-----------------|:----------:|:-----------:|:-------------:|:--------------:|
> > > | WaNet            | $y \to t$       |    56.78*   |    99.49*    |     57.05     |      86.98     |
> > > | LIRA             | $y \to y + 1$   |     58.    |     59.     |     49.16     |      99.94     |
> > > | Sleeper Agent    | $s \to t$       |   56.92*   |    6.00*    |     56.92     |      4.00      |
> > > | Label Consistent | $y \to t$       |   57.03*   |    9.84*    |     56.78     |      6.85      |
> > > | NARCISSUS        | $y \to t$       |    64.65   |    85.81    |     43.16     |      0.14      |
> > >
> > > It is notable that LIRA is unable
> > > to attain the claimed clean accuracies,
> > > and there are several [GitHub issues](https://github.com/sunbelbd/invisible_backdoor_attacks/issues) reporting difficulties
> > > in reproducing the results.
> > > As for NARCISSUS,
> > > we were unable to reproduce their nominal Tiny-ImageNet results,
> > > due to the lack of official hyper-parameter settings.
> > >
> > > We again thank the reviewer
> > > for providing an overall positive comment
> > > on our paper.
> > > Please feel free to let us know
> > > if you have more suggestions
> > > to further strengthen our paper.
> > >
> > > [1]: Wu et al., BackdoorBench: A Comprehensive Benchmark of Backdoor Learning, NeurIPS 2022.

---

> ### Author Response · Authors · 2022-12-05
> **Thanks for your reviews**
>
> Dear reviewer 2ChX,
>
> Thank you for your constructive comments and prompt feedback. We hope we have properly answered your questions and accommodated the feedback. All the feedback and discussions are helpful in improving the quality of our paper. Again, we appreciate your reviews and feedback on our response.
>
> Best,
>
> Paper 4225 authors.

---

### Author Response · Authors · 2022-11-17
**Detailed improvements to our paper.**

Dear reviewers,

We have improved the manuscript (updates are highlighted in blue) to strengthen its quality and address the remaining concerns. Besides the improvements mentioned in the initial replies to the reviewers, we summarize the additional refinements as follows:
* The introduction section now provides a clearer explanation of the motivation and novelty of Flareon.
* Table 5 of Section 4.5 has been updated with values either from BackdoorBench [1] or reproduced by us using the official implementations.
* Figure 6 of Appendix B has been updated to compare triggers added by label-consistent backdoor attacks (LC) [2].
* In Appendix C:
	* We include additional ablation of distribution types for trigger initialization (Table 9).
	* For a trained CIFAR-10 model, we further visualize the confusion matrix of clean image predictions and the ASR matrix of attack images evaluated on the test set.  The confusion matrix (Figure 8a) shows that Flareon does not noticeably impact clean accuracies of all labels. Moreover, the ASR matrix (Figure 8b) further visualizes the capabilities of *any2any* backdoors. *Any* images of *any* class can be attacked with *any* target-conditional triggers with very high success rates.
	* In Appendix C.1, we add defense experiments that employ randomized smoothing [3] and RAB [4], as requested by reviewers 2ChX and J6kD.
	* In Appendix C.2, we explain in depth the challenges of applying LC to our code-injection threat model, to clearly differentiate Flareon from LC. Figure 11 further compares the NC defense results on Flareon and LC, and Table 12 highlights the LC’s impact on clean accuracies.

As this completes our initial response, we hope you can kindly start **engaging in discussion** early on the remaining concerns that you may still have after reading our rebuttal. We look forward to an insightful discussion on our paper.

Best,

Paper 4225 authors.


[1]:  Wu et al., BackdoorBench: A Comprehensive Benchmark of Backdoor Learning, NeurIPS 2022.

[2]: Turner et al., Label-Consistent Backdoor Attacks, https://arxiv.org/abs/1912.02771.

[3]: Wang et al., On Certifying Robustness against Backdoor Attacks via Randomized Smoothing, https://arxiv.org/abs/2002.11750.

[4]: Weber et al., RAB: Provable Robustness Against Backdoor Attacks, S&P 2023.

---

> ### Author Response · Authors · 2022-11-30
> **Discussion reminder**
>
> Dear reviewers *t5Up* and *J6kD*,
>
> We still haven't heard from you following our updated manuscript and responses two weeks ago. This is a gentle reminder to both reviewers so that we have adequate time for a productive discussion.
>
> Thank you.

---

> > ### Author Response · Authors · 2022-12-10
> > **Approaching the end of discussion (Dec. 12).**
> >
> > Dear reviewers *t5Up* and *J6kD*,
> >
> > This is a gentle reminder that we are now approaching the end of the 2nd discussion period (December 12) in **2 days**, and yet we haven't heard back from you since our initial rebuttals to your reviews.
> >
> > We believe your concerns are addressed by our responses. We would appreciate if you can take a look at them and hope to have an insightful discussion on our paper.
> >
> > Thank you.

---

### Decision · Program_Chairs · 2023-01-20

**Decision:**

Reject

**Justification For Why Not Higher Score:**

See above

**Justification For Why Not Lower Score:**

N/A

**Metareview: Summary, Strengths And Weaknesses:**

This work studied a new threat model where the attacker could insert malicious code within the data augmentation module, and acquired no control over and no prior knowledge of the rest of the training algorithm, as well as the training data. It proposed a new type called any-to-any attack.

The reviewers had several important concerns about the novelty and technical contribution, some careless claims about existing works, inadequate literature reviews, insufficient experiments. Some of them are not well addressed in the rebuttal.

Besides, I also two additional concerns:
1. The practicality studied threat model should be discussed with more details. It is glad to see a new but really practical threat model in the community.
2. The authors claimed that 80% training sampled are affected to achieve high ASR, the clean accuracy is not affected much. That seems surprising.

Overall, exploring new threat model and new attack type is appreciated. However, the current shape of this work is not ready. Please keep improving this work.